# The impact of initial treatment strategy and survival time on quality of end-of-life care among patients with oesophageal and gastric cancer: A population-based cohort study

Karin Dalhammar [1,2][◉]*, Marlene Malmström[1,2][◉], Maria Schelin[1,3][◉], Dan Falkenback[3,4][◉], Jimmie Kristensson[1,2][◉]

1 Institute for Palliative Care, Lund University and Region Skåne, Lund, Sweden, 2 Department of Health Sciences, Faculty of Medicine, Lund University, Lund, Sweden, 3 Department of Clinical Sciences, Faculty of Medicine, Lund University, Lund, Sweden, 4 Department of Surgery, Skåne University Hospital, Lund, Sweden

◉ These authors contributed equally to this work.
* Karin.dalhammar@med.lu.se

**Data Availability Statement:** Data cannot be shared publicly because of regulations in the Swedish Data Protection Act (2018:218; 2019;

## Abstract

### Background

Oesophageal and gastric cancer are highly lethal malignancies with a 5-year survival rate of 15–29%. More knowledge is needed about the quality of end-of-life care in order to understand the burden of the illness and the ability of the current health care system to deliver timely and appropriate end-of-life care. The aim of this study was to describe the impact of initial treatment strategy and survival time on the quality of end-of-life care among patients with oesophageal and gastric cancer.

### Methods

This register-based cohort study included patients who died from oesophageal and gastric cancer in Sweden during 2014–2016. Through linking data from the National Register for Esophageal and Gastric Cancer, the National Cause of Death Register, and the Swedish Register of Palliative Care, 2156 individuals were included. Associations between initial treatment strategy and survival time and end-of-life care quality indicators were investigated. Adjusted risk ratios (RRs) with 95% confidence intervals were calculated using modified Poisson regression.

### Results

Patients with a survival of ≤3 months and 4–7 months had higher RRs for hospital death compared to patients with a survival ≥17 months. Patients with a survival of ≤3 months also had a lower RR for end-of-life information and bereavement support compared to patients with a survival ≥17 months, while the risks of pain assessment and oral assessment were not associated with survival time. Compared to patients with curative treatment, patients with no tumour-directed treatment had a lower RR for pain assessment. No significant

219) and Ethical Review Act (2003:460), data are available from the holders of the registers; NREV (Jan Johansson, Jan.johansson@med.lu.se) and SRPC (Maria Olsson, maria.olsson@regionkalmar.se) for researcher who meet the criteria for access to confidential data.

**Funding:** This work was supported by the ALF (governmental founding from the Swedish NHS; grant number ALF 2018:0092; URL:https://www.med.lu.se/intramed/styrning_organisation/ekonomi_alf/alf) and the Sjöberg Foundation (grant number Sjöberg F2019/210; URL: https://sjobergstiftelsen.se) awarded to JK. The funders had no role in study design, data collection and analysis, decision to publish, or preparation of the manuscript.

**Competing interests:** The authors have declared that no competing interests exist.

differences were shown between the treatment groups regarding hospital death, end-of-life information, oral health assessment, and bereavement support.

## Conclusions

Short survival time is associated with several indicators of low quality end-of-life care among patients with oesophageal and gastric cancer, suggesting that a proactive palliative care approach is imperative to ensure quality end-of-life care.

## Introduction

Oesophageal and gastric cancer are the 6th and 3rd leading causes of cancer mortality in the world [1]. In Sweden, about 1300 people are diagnosed with these cancers annually. The diagnosis is associated with late presentation and poor survival, and despite its relatively low incidence is responsible for about 1000 deaths each year in Sweden [2]. Due to the poor prognosis, a comprehensively accessible health care service and high-quality palliative care are paramount in order to ensure the best possible quality of life (QOL) for these patients.

Although the majority (75%) of patients diagnosed with oesophageal and gastric cancer are incurable [3, 4], research has primarily focused on anti-cancer treatment and the postoperative trajectory among patients treated with a curative intent. It is well known that curatively-intended surgery is associated with severe postoperative complications and deterioration in QoL, both in the short-term and the long-term perspective [5, 6]. In order to properly understand the burden of illness and the ability of the current health care system to deliver timely and appropriate palliative care to patients with oesophageal and gastric cancer, we need more knowledge about the quality of end-of-life (EOL) care among these patients.

The prognosis of oesophageal and gastric cancer is poor, with an overall 5-year survival rate of 15–29% [7]. Surgery alone or in combination with neoadjuvant therapy is the mainstay treatment for cure [8, 9], but due to comorbidities and/or late-stage disease at presentation only 25% of patients are considered suitable for curatively-intended treatment [3, 4]. The prognosis for patients undergoing potentially curative surgery is also poor, with a recurrence rate of 30–67% in the first postoperative year [10, 11]. For patients who are not considered suitable for curatively-intended surgery, the main treatment is chemotherapy aimed at prolonging survival, maintaining QOL, and relieving symptoms [4, 12]. Regardless of whether the initial treatment strategy is curative or palliative, patients experience emotional distress, a severely reduced QOL and a range of diagnosis-specific and treatment-related problems and side effects such as difficulties with nutrition or elimination [13–15]. Considering the poor survival rate and the multiple symptom burden, a palliative care approach during the entire course of illness is of utmost importance.

Palliative care is defined by the World Health Organization as an approach which aims to mitigate suffering and optimize QOL among patients and their families facing problems associated with life-threatening illness [16]. The American College of Surgeons has recommended that palliative care should be integrated early in the course of the disease, concurrently with active treatment [17]. Palliative care of patients who are nearing EOL is often described using the term "EOL care". The American Society of Clinical Oncology (ASCO) has suggested a number of quality indicators for EOL care: no more than one emergency department (ED) visit during the last month of life, pain assessment, not dying in the hospital, no intensive care unit admission during the last 30 days of life, enrolment in hospice for a meaningful length of

time, and no chemotherapy administered within the last 2 weeks of life [18]. Several studies have shown inadequacies in the quality of EOL care, characterized by late delivery [19], unplanned hospitalization [20–22], and aggressive care such as non-beneficial medical treatments or interventions [23]; however, this differs substantially across cancer diagnoses [24]. One study indicated that surgically-treated patients receive significantly less hospice care in the last year of life compared with medically-treated patients [25], while another demonstrated that patients undergoing resection are less likely to enrol in hospice early and more likely to be admitted to an acute-care hospital in the last month of life [26].

Patients with oesophageal and gastric cancer often suffer from a combination of physical and emotional symptoms which may increase as the disease progress. The rapid disease deterioration and the complex needs of these patients pose a significant challenge for health care providers in ensuring timely high-quality EOL care. However, research focusing on this patient group from a palliative care perspective is scarce, despite extensive evidence regarding unmet EOL quality indicators. One Korean study reported that 30% of patients with gastric cancer received chemotherapy within a month of death and 39% visited the ED more than once during the final month of life [27]. A French study found that 77% of patients with advanced oesophageal and gastric cancer died in hospital, and that 8% of these patients received chemotherapy during the final week of life [28]. Treatment with surgery [25] or chemotherapy [29] is associated with underuse of palliative care in terms of palliative care consultation [30, 31] and late hospice referral [32]. This indicates that treatment characteristics may influence the quality of EOL care, and should thus be taken into account in order to understand barriers to high quality of care.

Given the poor survival rate among patients with oesophageal and gastric cancer and the increasing policy attention to early integrated palliative care, there is a great need for studies examining the potential impact of treatment characteristics and survival time on quality of EOL care. Such knowledge could provide valuable insight into the ability of current health care to deliver timely and appropriate EOL care, as well as guiding resource allocation and informing future interventions aimed at improvement of EOL care. The aim of this study was therefore to describe the impact of initial treatment strategy and survival time on the quality of EOL care among patients with oesophageal and gastric cancer.

## Methods

### Design

This was a population-based observational cohort study.

### Study population

The sample comprised 2636 individuals who died in Sweden between 1 January 2014 and 31 December 2016 with oesophageal and gastric cancer as the underlying cause of death. They were identified by means of three registers: the National Register for Esophageal and Gastric Cancer (NREV), the National Cause of Death Register, and the Swedish Register of Palliative Care (SRPC).

### Data collection

Data were collected from the three registers. NREV is a national quality register comprising information about diagnostics, clinical manifestations, outcome of surgical treatment, and follow-up of oesophageal and gastric cancer. The national completeness is more than 95% [33]. NREV was used both to identify relevant patients and to obtain data about date of diagnosis,

tumour site, histology, performance status according to the Eastern Cooperative Oncology Group scale (0–5, with lower values representing better function) [34], clinical M-stage (M1: the cancer has metastasized, M0: no metastasis), and whether the initial treatment strategy was curative (tumour-directed treatment such as surgery /chemotherapy/radiotherapy with a curative intent), palliative (tumour-directed treatment such as surgery/chemotherapy/radiotherapy with a palliative intent), or no tumour-directed therapy.

The National Cause of Death Register is held by the National Board of Health and Welfare and covers 99% of all deaths in Sweden [35]. Data about date of death and underlying cause of death according to the Tenth Revision of the International Statistical Classification of Diseases and Related Health Problems (ICD-10) were extracted from this register and linked to NREV. Only individuals with oesophageal and gastric cancer as the underlying cause of death were included.

SRPC is a national quality register focusing on the care of patients during their final week of life, regardless of diagnosis, place of death, or level of care. Data are collected via an EOL questionnaire which is completed by health care staff after the patient's death. The items in this questionnaire are based on the principles of a good death proposed by the British Geriatric Society [36]. SRPC includes information about several aspects of quality of EOL care, such as place of death, symptom assessment, prescription of pro re nata drugs for common symptoms, and information provided to patient and next of kin about transition to EOL care [37]. The completeness of SRPC for patients with cancer is 86% [38]. Data about place of death, pain assessment (whether a visual analogue scale/numeric rating scale was used for pain evaluation), information provided to the patient about transition to EOL care (EOL information), oral health assessment, and whether the next of kin were offered bereavement support were extracted and linked to the corresponding patient.

In the linked dataset, 443 persons were not registered in SRPC and so could not be included in the analyses. Another 17 lacked information about initial treatment strategy from NREV, and 20 were registered with unexpected death in SRPC, leading to missing data. These exclusions left a total of 2156 persons for inclusion in the analysis (Fig 1).

## Data analysis

The sample (n = 2156) was categorized into three pre-defined categories according to the initial treatment strategy: curative, palliative, and no tumour-directed treatment.

Pain assessment, EOL information, oral health assessment, and bereavement support were reported as "yes", "no", or "don't know". Answers of "don't know" were excluded, leaving a binary categorization of "yes" or "no" for each measure. Place of death was reported as "resident care facility–permanent stay", "resident care facility–short term stay", "hospital ward, without palliative specialization", "palliative inpatient care unit", "own home with support from advanced home care", "own home with support from basic home care", or "other". These answers were also categorized in binary terms, as "yes" or "no" for hospital death (hospital ward, without palliative specialization). For each analysis, patients with unknown exposure or outcome were excluded.

Baseline data on demographic and clinical characteristics were analysed with descriptive and analytical statistics. Differences in mean and proportion were calculated by using ANOVA for numerical data and the chi-square test and Fisher's exact test for nominal data.

Modified Poisson regression was used to calculate risk ratios (RRs) with 95% confidence intervals (CI) to assess the impact of initial treatment strategy on each quality of EOL care outcome. The category "curative" was used as reference category. Several possible confounders

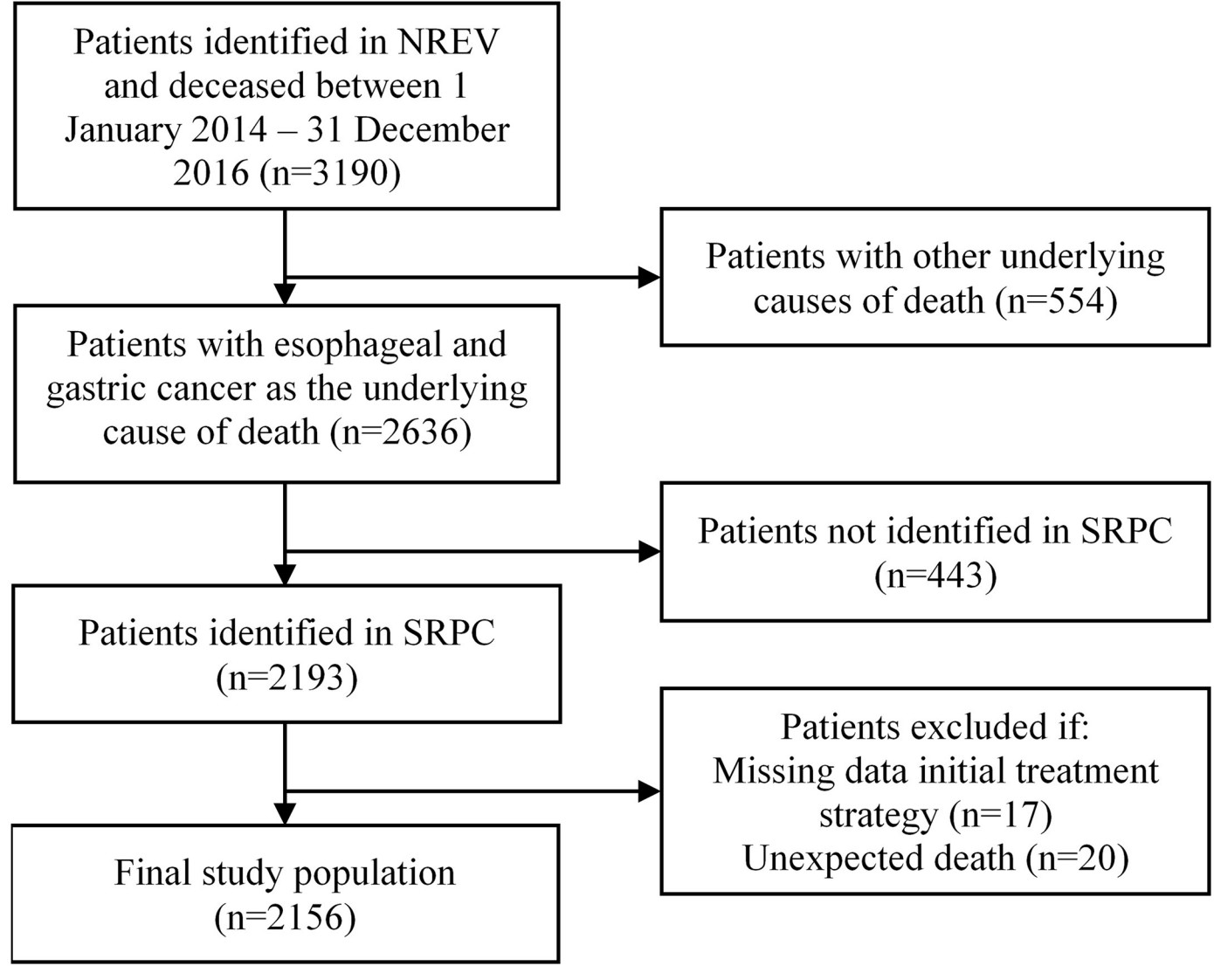

**Fig 1. Derivation of final study population.**

were taken into account in the statistical models: sex, age (categorized into quartiles of ≤65, 66–72, 73–79, and ≥80), M-stage, and performance status.

Modified Poisson regression was also used to assess the associations between survival time and each quality of EOL care outcome. The sample was stratified according to survival time quartiles: ≤3, 4–7, 8–17, and ≥17 months. RR for each survival range was calculated by using long-term survivors (≥17 months) as reference category.

To account for potential misclassification and to assess the robustness of the results from the primary analysis, a sensitivity analysis was performed by classifying patients initially intended for a curative treatment strategy according to whether the planned curative treatment was given or not. Model adjustment was similar to the primary analyses.

All statistical analyses were performed using version 25 of IBM SPSS Statistics. A p-value of <0.05 was used to define statistical significance.

### Ethical approval and ethical considerations

The study was approved by the Lund Regional Ethics Review Board (REC number: 2018/03, 2018/270). Informed consent was not obtained because all patients were deceased.

## Results

Of the 2156 individuals included, 1459 (68%) were men. The mean age at diagnosis was 71 years (SD ±11.7) (Table 1). Adenocarcinoma and squamous cell carcinoma accounted for 78% and 15% of all cancers, respectively. Among all patients, 1129 (52.4%) had a tumour originating in the oesophagus, 1027 (42%) had distant metastases (M1) at the time of diagnosis, and 749 (36%) had a performance status score of 1. In total, 1074 (50%) received a palliative treatment strategy, 721(33%) a curative treatment strategy, and 361 (17%) no tumour-directed treatment. In terms of survival, 679 (31.6%) survived ≤3 months, 405 (18.8%) survived 4–7 months, 555 (25.8%) survived 8–16 months, and 510 (23.7%) survived ≥17 months.

### Quality of end-of-life care by initial treatment strategy

Compared to patients with a curative treatment, patients with no tumour-directed treatment had a significantly lower adjusted RR of having pain assessment (RR 0.83, 95% CI 0.70–0.99) while patients with palliative treatment had similar risk to the curative group (RR 0.91, 95% CI 0.80–1.02). There were no differences between the groups in terms of hospital death, EOL information, oral health assessment, or bereavement support (Table 2). The findings from the sensitivity analyses confirmed that there was no systematic effect of treatment, either planned nor given, on quality of EOL care.

### Quality of end-of-life care by survival time

Compared to the long-term survivors (≥17 months), patients who survived ≤3 months and 4–7 months had significantly higher adjusted RRs for hospital death (RR 2.36; 95% CI 1.85–3.00 and RR 1.60; 95% CI 1.23–2.07 respectively) (Table 3). Patients who survived ≤3 months also had significantly lower RRs for EOL information (RR 0.94; 95% CI 0.88–0.99) and bereavement support (RR 0.93; 95% CI 0.87–0.98) compared to patients who survived ≥17 months; no corresponding differences were seen for patients surviving 4–7 months. There were no differences between any of the groups in terms of oral health assessment or pain assessment.

## Discussion

This population-based cohort study of patients with oesophageal and gastric cancer showed that short-term survivors are more likely to receive poorer quality of EOL care for three out of five examined outcomes when compared to long-term survivors. Patients with a survival of <7 months had a higher likelihood of hospital death, and patients with a survival of <3 months also had a lower likelihood of EOL information and bereavement support compared to patients with a survival ≥17 months. However, no systematic significant differences in quality of EOL care were observed between patients with palliative treatment or no tumour-directed treatment and patients with curative treatment.

Whereas previous studies have examined the association between EOL care and treatments given in the months before death [32], to our knowledge this is the first study to investigate how primary treatment *strategy* relates to quality of EOL care. We found that patients who received palliative treatment or no tumour-directed treatment had the same likelihood of EOL information, bereavement support, oral health assessment, and hospital death as patients with

**Table 1. Baseline demographic and clinical characteristics by initial treatment strategy and survival time.**

| | Total n = 2156 | Initial treatment strategy | | | | Survival time (months) | | | | |
|---|---|---|---|---|---|---|---|---|---|---|
| | | Curative n = 721 (33.4%) | Palliative n = 1074 (49.8%) | No tumour-directed treatment n = 361 (16.7%) | p-value | ≤3 n = 679 (31.6%) | 4–7 n = 405 (18.8%) | 8–16 n = 555 (25.8%) | ≥17 n = 510 (23.7%) | p-value |
| Age (mean±sd) | 71.2±11.7 | 67.6±10.5 | 71.0±11.7 | 78.8±10.2 | 0.000 | 74.8±11.0 | 72.1±11.9 | 69.6±11.2 | 67.3±11.5 | 0.000 |
| Men, n (%) | 1459 (68) | 500 (69.3) | 746 (69.5) | 213 (59) | 0.001 | 446 (65.7) | 263 (64.9) | 401 (72.3) | 345 (67.6) | 0.047 |
| Women, n (%) | 697 (32) | 221 (30.7) | 328 (30.5) | 148 (41) | | 233 (34.3) | 142 (35.1) | 154 (27.7) | 165 (32.4) | |
| **Survival time from diagnosis** | 2149[1] | | | | | | | | | |
| Months (mean, sd) | 12.2±15.1 | 21.8±19.6 | 8.0±8.4 | 5.4±10.1 | 0.000 | | | | | |
| **Histological type, n (%)** | 2141[2] | | | | 0.000 | | | | | 0.030 |
| Adenocarcinoma | 1667 (77.9) | 559 (77.7) | 819 (76.8) | 289 (81.4) | | 524 (78.0) | 297 (73.5) | 429 (77.9) | 413 (81.3) | |
| Non differentiated | 31 (1.4) | 7 (1.0) | 18 (1.7) | 6 (1.7) | | 13 (1.9) | 4 (1.0) | 11 (2.0) | 3 (0.6) | |
| Squamous cell carcinoma | 329 (15.4) | 120 (16.7) | 180 (16.9) | 29 (8.2) | | 94 (14.0) | 77 (19.1) | 91 (16.5) | 65 (12.8) | |
| Others | 114 (5.3) | 33 (4.6) | 50 (4.7) | 31 (8.7) | | 41 (6.1) | 26 (6.4) | 20 (3.6) | 27 (5.3) | |
| **M-stage at diagnosis, n (%)** | 2135[3] | | | | 0.000 | | | | | 0.000 |
| M0 | 1227 (57.5) | 686 (95.9) | 383 (35.9) | 158 (44.6) | | 237 (35.4) | 224 (56.1) | 354 (63.9) | 411 (81.4) | |
| M1 | 908 (42.5) | 29 (4.1) | 683 (64.1) | 196 (55.4) | | 433 (64.6) | 175 (43.9) | 200 (36.1) | 94 (18.6) | |
| **Performance status, n (%)** | 2063[4] | | | | 0.000 | | | | | 0.000 |
| 0 | 559 (27.1) | 333 (48.3) | 211 (20.5) | 15 (4.4) | | 61 (9.4) | 85 (22.3) | 189 (35.4) | 223 (45.7) | |
| 1 | 749 (36.3) | 279 (40.5) | 398 (38.6) | 72 (21.0) | | 178 (27.3) | 133 (34.8) | 226 (42.3) | 210 (43.0) | |
| 2 | 509 (24.7) | 67 (9.7) | 310 (30.1) | 132 (38.5) | | 230 (35.3) | 131 (34.3) | 98 (18.4) | 47 (9.6) | |
| 3 | 208 (10.1) | 10 (1.5) | 104 (10.1) | 94 (27.4) | | 151 (23.1) | 31 (8.1) | 19 (3.6) | 6 (1.2) | |
| 4 | 38 (1.8) | 0 (0) | 8 (0.8) | 30 (8.7) | | 32 (4.9) | 2 (0.5) | 2 (0.4) | 2 (0.4) | |
| **Site of primary tumour** | 2156 | | | | 0.000 | | | | | 0.001 |
| Oesophageal cancer | 1129 (52.4) | 392 (54.4) | 622 (57.9) | 115 (31.9) | | 324 (47.7) | 228 (56.3) | 320 (57.7) | 252 (49.4) | |
| Gastric cancer | 1027 (47.6) | 329 (45.6) | 452 (42.1) | 246 (68.1) | | 355 (52.3) | 177 (43.7) | 235 (42.3) | 258 (50.6) | |

[1] 7 missing;

[2] 15 missing;

[3] 21 missing;

[4] 93 missing

a curative treatment. This suggests that even though there are many differences between the three categories, in terms of goals of care, medical treatment, and interventions, these differences do not appear to influence quality of EOL care. Our findings are in contrast to previous reports that tumour-directed treatment given in the months before death, such as surgery [25] and chemotherapy [29], is associated with inferior quality of EOL care in terms of unplanned hospitalization in the last month of life [26] and late hospice referral [32]. There are several plausible explanations for the divergent findings in the present study. First, the initial treatment characteristics by which the three study groups were categorized were different compared to those in other studies. In this study, the palliative and curative treatment groups included both surgically- and chemotherapy-treated patients, whereas in the study by Wu et al. [29] patients were categorized as receiving or not receiving chemotherapy. This means

**Table 2. Quality of end-of-life care by initial treatment strategy.**

| | Hospital death | | | |
| --- | --- | --- | --- | --- |
| | Unadjusted | | Adjusted* | |
| **Initial treatment** | RR (95% CI) | p-value | RR (95% CI) | p-value |
| Not tumour-directed | 1.03 (0.84–1.27) | 0.78 | 1.11 (0.83–1.48) | 0.47 |
| Palliative | 0.92 (0.78–1.08) | 0.31 | 0.89 (0.72–1.11) | 0.32 |
| Curative | 1.00 (reference) | | 1.00 (reference) | |
| | **Pain assessment** | | | |
| | Unadjusted | | Adjusted* | |
| **Initial treatment** | RR (95% CI) | p-value | RR (95% CI) | p-value |
| Not tumour-directed | **0.82 (0.73–0.94)** | **0.003** | **0.83 (0.70–0.99)** | **0.04** |
| Palliative | 0.92 (0.84–1.00) | 0.06 | 0.91 (0.80–1.02) | 0.11 |
| Curative | 1.00 (reference) | | 1.00 (reference) | |
| | **EOL information** | | | |
| | Unadjusted | | Adjusted* | |
| **Initial treatment** | RR (95% CI) | p-value | RR (95% CI) | p-value |
| Not tumour-directed | 0.96 (0.91–1.02) | 0.20 | 0.99 (0.92–1.06) | 0.72 |
| Palliative | 1.01 (0.97–1.05) | 0.58 | 1.02 (0.97–1.07) | 0.44 |
| Curative | 1.00 (reference) | | 1.00 (reference) | |
| | **Oral health assessment** | | | |
| | Unadjusted | | Adjusted* | |
| **Initial treatment** | RR (95% CI) | p-value | RR (95% CI) | p-value |
| Not tumour-directed | 0.91 (0.81–1.01) | 0.07 | 0.99 (0.86–1.14) | 0.90 |
| Palliative | 0.94 (0.87–1.01) | 0.09 | 0.97 (0.87–1.07) | 0.55 |
| Curative | 1.00 (reference) | | 1.00 (reference) | |
| | **Bereavement support** | | | |
| | Unadjusted | | Adjusted* | |
| **Initial treatment** | RR (95% CI) | p-value | RR (95% CI) | p-value |
| Not tumour-directed | 0.96 (0.91–1.02) | 0.16 | 0.95 (0.88–1.02) | 0.15 |
| Palliative | 1.01 (0.97–1.04) | 0.78 | 0.99 (0.94–1.04) | 0.65 |
| Curative | 1.00 (reference) | | 1.0 (reference) | |

*Adjusted for age, sex, M-stage, and performance status.

that potential differences in quality of EOL care with regard to specific treatment methods (surgery vs. chemotherapy) cannot be distinguished in our study.

Quality of EOL care is associated with access to palliative care consultation and support throughout the cancer trajectory [39]. The differences between the results of the current study and previous research must therefore be considered in the light of palliative care access. Previous studies [25, 30–32] have been conducted in the USA, where the model of palliative-care delivery and policies for hospice referral differ from those in Sweden. For instance, in the USA, life-prolonging treatment with chemotherapy or radiation therapy precludes patients from hospice service and there is only limited availability of palliative care for patients living in the community who are not hospice eligible [40]. There are no such restrictions in Sweden [41], and so patients with and without tumour-directed therapy are equally likely to receive these services.

In order to standardize and enhance the quality of care, Swedish guidelines state that all patients diagnosed with oesophageal and gastric cancer should be discussed at a multidisciplinary team conference throughout the cancer trajectory. Multidisciplinary team meetings

**Table 3. Quality of end-of-life care by survival time.**

| | Hospital death | | | |
| --- | --- | --- | --- | --- |
| | Unadjusted | | Adjusted* | |
| Survival time (months) | RR (95% CI) | p-value | RR (95% CI) | p-value |
| ≤3 | **1.70 (1.40–2.10)** | **0.000** | **2.36 (1.85–3.00)** | **0.000** |
| 4–7 | **1.30 (1.04–1.68)** | **0.023** | **1.60 (1.23–2.07)** | **0.000** |
| 8–16 | 1.10 (0.88–1.40) | 0.378 | 1.19 (0.93–1.53) | 0.161 |
| ≥17 | 1.00 (reference) | | 1.00 (reference) | |
| | **Pain assessment** | | | |
| | Unadjusted | | Adjusted* | |
| Survival time (months) | RR (95% CI) | p-value | RR (95% CI) | p-value |
| ≤3 | **0.88 (0.79–0.98)** | **0.023** | 0.90 (0.78–1.03) | 0.126 |
| 4–7 | 0.92 (0.81–1.04) | 0.171 | 0.92 (0.80–1.05) | 0.224 |
| 8–16 | 1.03 (0.93–1.14) | 0.591 | 1.04 (0.93–1.16) | 0.507 |
| ≥17 | 1.00 (reference) | | 1.00 (reference) | |
| | **EOL information** | | | |
| | Unadjusted | | Adjusted* | |
| Survival time (months) | RR (95% CI) | p-value | RR (95% CI) | p-value |
| ≤3 | **0.96 (0.91–1.0)** | **0.049** | **0.94 (0.88–0.99)** | **0.036** |
| 4–7 | 1.00 (0.95–1.0) | 0.842 | 1.00 (0.94–1.04) | 0.841 |
| 8–16 | 0.99 (0.95–1.0) | 0.750 | 0.99 (0.94–1.03) | 0.577 |
| ≥17 | 1.00 (reference) | | 1.00 (reference) | |
| | **Oral health assessment** | | | |
| | Unadjusted | | Adjusted* | |
| Survival time (months) | RR (95% CI) | p-value | RR (95% CI) | p-value |
| ≤3 | **0.87 (0.79–0.96** | **0.004** | 0.91 (0.81–1.01) | 0.092 |
| 4–7 | **0.89 (0.80–0.99)** | **0.031** | 0.90 (0.80–1.00) | 0.063 |
| 8–16 | 0.98 (0.89–1.07) | 0.664 | 0.98 (0.89–1.07) | 0.609 |
| ≥17 | 1.00 (reference) | | 1.00 (reference) | |
| | **Bereavement support** | | | |
| | Unadjusted | | Adjusted* | |
| Survival time (months) | RR (95% CI) | p-value | RR (95% CI) | p-value |
| ≤3 | **0.95 (0.91–0.99)** | **0.028** | **0.93 (0.87–0.98)** | **0.008** |
| 4–7 | 0.96 (0.91–1.01) | 0.139 | 0.95 (0.90–1.00) | 0.070 |
| 8–16 | 1.00 (0.97–1.06) | 0.542 | 1.00 (0.96–1.05) | 0.858 |
| ≥17 | 1.00 (reference) | | 1.00 (reference) | |

*Adjusted for age, sex, M-stage, and performance status.

have been shown to facilitate re-evaluation of treatment goals, to improve quality of palliative care by increasing the rate of referral to palliative care services [42], and to decrease the risk of ED visits [43]. It is possible that patients initially intended for curative treatment but whose disease was unresponsive to this approach could have been switched to palliative treatment along the care trajectory. Thus, patients who were initially regarded as curative could have obtained similar access to quality EOL care as patients who were initially regarded as palliative. For example, Van den Block et al. demonstrated that a curative treatment goal during the entire last three months of life increased the odds of hospitalization by five times compared to patients with a palliative goal during the entire period, but that the odds of hospitalization

decreased among patients whose treatment goal was changed from curative to palliative during the last month of life [44].

The finding of the present study that there was no association between primary treatment strategy and quality of EOL care suggests that the quality of EOL care is not affected by tumour-directed treatment given in the *early* stage of the care trajectory. There might, how-ever, be an impact from treatment given at a *later* stage in the disease course, as indicated by previous research in other settings [32]. It remains to be elucidated whether the impact of treatment on quality of EOL care is in fact time-varying.

There were significant differences in quality of EOL care between short-term survivors and long-term survivors. Patients with a survival of <7 months, which corresponds to 50% of our cohort, had a risk of hospital death more than twice that of patients with a survival of ≥17 months. This indicates that time since diagnosis is an important factor for quality of EOL care, in line with earlier research [45–47]. Kelly et al. showed that patients with lung cancer who died within 30 days of diagnosis were more likely to die in hospital compared to long-term survivors [48], and Brooks el al. found a negative association between survival time and in-hospital death, with patients surviving <6 months having the greatest risk [46]. Although some hospitalizations might be necessary, and might benefit patients during the EOL, hospitalization is considered suboptimal with regard to quality of EOL care [49] and the hospital is reported to be the least preferred location of death among patients [50]. A pos-sible explanation for our finding could be that patients who die quickly after diagnosis have limited time for advanced care planning and multidisciplinary care coordination for smooth transfer from hospital to other care settings. Previous research has shown that advanced care planning is an important factor for quality of EOL care in terms of decreasing in-hospital death and increasing hospice referral [51]. However, Prater et al. demonstrated that advanced care planning at an early stage (≥30 days before death) was more strongly associated with quality care outcomes compared to planning that occurred near death [52]. The availability and readiness of current health care services might not be sufficient to meet the quality stan-dard for place of death when time is scarce. Considering the poor prognosis for the majority of patients with oesophageal and gastric cancer, a proactive care approach with advanced care planning at an early stage would seem crucial in order to decrease the risk of in-hospital death.

EOL communication could also give patients and their families a more realistic understand-ing of the prognosis, and encourage attention to practical arrangements according to their preferences regarding care and place of death. However, the present study showed that short-term survivors were less likely than long-term survivors to have EOL communication. There are several possible explanations for this, both from the perspective of the patient and from that of the physician. Physician-related and patient-related factors such as prognostic uncer-tainty and difficulty accepting a poor prognosis have previously been identified as barriers to initiating EOL discussion [53]. It may be more difficult to admit a poor prognosis when a patient is newly diagnosed. Physicians also tend to overestimate survival among terminally ill patients [54], and it is likely that a rapid decline can make accurate prediction even more diffi-cult. As a consequence, EOL communication may be initiated late or not at all. The results of the current study imply that neither patients nor health care professionals are prepared when death comes quickly after a diagnosis of oesophageal and gastric cancer. This lack of EOL dis-cussion is a missed opportunity for health care professionals to elicit the patient's values and preferences, and hence to tailor the care accordingly. Previous studies have shown an associa-tion between EOL discussions and fewer in-hospital deaths and aggressive interventions at EOL [55]. However, such discussions need to be held early in the disease trajectory in order to

have a beneficial impact on the quality of EOL care. A study by Gieniusz et al. indicated that each additional day from hospital admission to EOL communication increased the risk of death as an inpatient by 4% [55].

This study has several strengths. The cohort was population-based and included a relatively large sample with 2156 patients from different geographical areas, representative for Sweden. The study also covered EOL care provided at multiple health care settings, and may therefore provide a realistic picture of the quality of EOL care among patients with oesophageal and gastric cancer in Sweden. Our results were based on data from NREV, which has an accuracy of 91.1% and completeness rate of 95.5% among diagnosed patients [33]. The accuracy of SRPC has been shown to vary between items in the EOL questionnaire, but all the items included here had an accuracy of 85–95% [56].

There are also a few limitations of the present study. The categorization of patients was based on *planned* treatment, and some of the patients might not have been treated according to the initial plan. This particularly concerns patients with a curative intent who progressed into advanced disease after the initial treatment decision had been made. To account for potential misclassification and to assess the robustness of the results from the primary analysis, we performed a sensitivity analysis by reclassifying patients with an initially curative treatment strategy according to whether the planned curative treatment was given or not. However, we were unable to reclassify the other categories (palliative with tumour-directed treatment and no tumour-directed treatment) due to lack of follow-up data on the treatment given. Nevertheless, the findings from the sensitivity analysis were consistent with those from the primary analyses, suggesting that our findings are unlikely to have been driven by misclassification.

The study was also limited to deceased patients registered in SRPC, and EOL care characteristics may have been different for those not included. For example, it is possible that heath care providers with a low interest in EOL care are more likely to neglect SRPC registration. If there was any differential loss to follow-up between the study groups, the current study may underestimate association. However, potential errors in outcome classification are most likely unrelated to treatment/survival time, and therefore unlikely to be an important threat to the validity of this study. The analysis was adjusted for age, sex, M-stage, and performance status, but as we used an observational design, unmeasured confounding cannot be completely excluded.

In conclusion, the present study showed no association between initial treatment strategy and quality of EOL care among patients with oesophageal and gastric cancer. However, a short survival time was associated with several indicators of low-quality EOL care. This result has important implications for health care providers as well as patients with oesophageal and gastric cancer and their families. Our findings underscore the importance of a proactive palliative care approach with early EOL discussion, advanced care planning, and timely delivery of care in treating patients with oesophageal and gastric cancer. Great efforts have been made to implement standardized cancer pathways to reduce lead times for diagnosis and treatment of oesophageal and gastric cancer, and now it is time to extend these efforts in order to ensure high-quality EOL care.

## Acknowledgments

The authors gratefully acknowledge The National Register for Esophageal and Gastric Cancer (NREV) and The Swedish Register of Palliative Care (SRPC) for providing data and Dr. Jan Johansson for support.

## Author Contributions

**Conceptualization:** Karin Dalhammar, Marlene Malmström, Maria Schelin, Dan Falkenback, Jimmie Kristensson.

**Data curation:** Karin Dalhammar, Marlene Malmström, Maria Schelin, Dan Falkenback, Jimmie Kristensson.

**Formal analysis:** Karin Dalhammar, Marlene Malmström, Maria Schelin, Dan Falkenback, Jimmie Kristensson.

**Funding acquisition:** Jimmie Kristensson.

**Investigation:** Karin Dalhammar, Marlene Malmström, Maria Schelin, Dan Falkenback, Jimmie Kristensson.

**Methodology:** Karin Dalhammar, Marlene Malmström, Maria Schelin, Dan Falkenback, Jimmie Kristensson.

**Project administration:** Karin Dalhammar, Marlene Malmström, Maria Schelin, Dan Falkenback, Jimmie Kristensson.

**Resources:** Karin Dalhammar, Marlene Malmström, Maria Schelin, Dan Falkenback, Jimmie Kristensson.

**Software:** Karin Dalhammar, Marlene Malmström, Maria Schelin, Dan Falkenback, Jimmie Kristensson.

**Supervision:** Marlene Malmström, Maria Schelin, Dan Falkenback, Jimmie Kristensson.

**Validation:** Karin Dalhammar, Marlene Malmström, Maria Schelin, Dan Falkenback, Jimmie Kristensson.

**Visualization:** Karin Dalhammar, Marlene Malmström, Maria Schelin, Dan Falkenback, Jimmie Kristensson.

**Writing – original draft:** Karin Dalhammar, Marlene Malmström, Maria Schelin, Dan Falkenback, Jimmie Kristensson.

**Writing – review & editing:** Karin Dalhammar, Marlene Malmström, Maria Schelin, Dan Falkenback, Jimmie Kristensson.

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
