## [Decision Letter · Decision Letter 0]

9 Jun 2020

The impact of initial treatment strategy and survival time on quality of end-of-life care among patients with oesophageal and gastric cancer: A population-based cohort study

PONE-D-20-10092

Dear Dr. Dalhammar,

We are pleased to inform you that your manuscript has been judged scientifically suitable for publication and will be formally accepted for publication once it complies with all outstanding technical requirements.

With kind regards,

Jason Chia-Hsun Hsieh, M.D. Ph.D

Academic Editor

PLOS ONE

Journal Requirements:

1. We note that you have included the phrase “data not shown” in your manuscript. Unfortunately, this does not meet our data sharing requirements. PLOS does not permit references to inaccessible data. We require that authors provide all relevant data within the paper, Supporting Information files, or in an acceptable, public repository. Please add a citation to support this phrase or upload the data that corresponds with these findings to a stable repository (such as Figshare or Dryad) and provide and URLs, DOIs, or accession numbers that may be used to access these data. Or, if the data are not a core part of the research being presented in your study, we ask that you remove the phrase that refers to these data.

Additional Editor Comments (optional):

The study is well-designed, well-conducted, and well-described.

Reviewers' comments:

Reviewer's Responses to Questions

**Comments to the Author**

1. Is the manuscript technically sound, and do the data support the conclusions?

Reviewer #1: Yes

Reviewer #2: Yes

2. Has the statistical analysis been performed appropriately and rigorously? 

Reviewer #1: Yes

Reviewer #2: Yes

3. Have the authors made all data underlying the findings in their manuscript fully available?

Reviewer #1: No

Reviewer #2: Yes

4. Is the manuscript presented in an intelligible fashion and written in standard English?

Reviewer #1: Yes

Reviewer #2: Yes

5. Review Comments to the Author

Reviewer #1: This manuscript present an original point of view on the quality of end-of-life care based on survival time.

The statistical analysis are performed rigorously.

The subject is particulary difficult to translate in study due to the death of the subjects and the variability of the country habits about the end of life.

The main information we can retain about this article is to be pro-active about the supportive care as soon as possible

Reviewer #2: Interesting paper. I would be a bit more bold in conclusion that palliative care consultation should be done in first 30 days after diagnosis gastric or esophageal cancer to influence better outcomes.

6. PLOS authors have the option to publish the peer review history of their article (what does this mean?). If published, this will include your full peer review and any attached files.

Reviewer #1: No

Reviewer #2: No

---

## [Editor Report · Acceptance letter]

12 Jun 2020

PONE-D-20-10092 

The impact of initial treatment strategy and survival time on quality of end-of-life care among patients with oesophageal and gastric cancer: A population-based cohort study 

Dear Dr. Dalhammar:

I'm pleased to inform you that your manuscript has been deemed suitable for publication in PLOS ONE. Congratulations! Your manuscript is now with our production department. 

Kind regards, 

on behalf of

Dr. Jason Chia-Hsun Hsieh 

Academic Editor

PLOS ONE